

# Application of indole-3-butyric acid (IBA) enhances agronomic, physiological and antioxidant traits of *Salvia fruticosa* under saline conditions: a practical approach

Uğur Tan

Department of Field Crops, Aydin Adnan Menderes University, Aydin, Türkiye

## ABSTRACT

**Background:** Salinity stress is a significant challenge in agriculture, particularly in regions where soil salinity is increasing due to factors such as irrigation practices and climate change. This stress adversely affects plant growth, development, and yield, posing a threat to the cultivation of economically important plants like *Salvia fruticosa*. This study aims to evaluate the effectiveness by proactively applying indole-3-butyric acid (IBA) to *Salvia fruticosa* cuttings as a practical and efficient method for mitigating the adverse effects of salinity stress.

**Methods:** The factors were arranged as three different IBA doses (0, 1, and 2 g/L) and four different salinity concentrations (0, 6, 12, and 18 dS/m) in controlled greenhouse conditions. Plant height (PH), flower spike length (FSL), fresh shoot length (FRL), root length (RL), fresh root weight (FRW), fresh shoot weight (FSW), dried root weight (DRW), dried shoot weight (DSW), root/shoot index, drog (g/plant), relative water content (RWC), relative membrane permeability (RMP), chlorophyll content (SPAD), extraction yield (%), DPPH (2,2-Diphenyl-1-picrylhydrazyl), phenol content, flavonoid content, and ABTS (2,2′-Azino-bis (3-ethylbenzothiazoline-6-sulfonic acid)) values were measured.

**Results:** The results show that as salinity doses increased, all parameters showed a decline. However, with a one-time IBA application to the plant cuttings before the rooting stage, particularly at a concentration of 2 g/L, was effective for mitigating the negative effects of salinity stress. Across all measured parameters, IBA significantly reduced the adverse impacts of salinity on *Salvia fruticosa*.

## INTRODUCTION

*Salvia fruticosa* Miller (syn. *S. triloba*), commonly known as Anatolian sage and Greek sage, is native to both Türkiye and Greece (*Baytop, 1999*; *Papafotiou et al., 2021*). The leaves of *S. fruticosa* are traditionally used to make herbal tea. This plant is valued for its medicinal properties, including benefits for respiratory tract infections, neurological disorders, and diarrhea, as well as its pain-relieving effects (*Baytop, 1999*). Species within the *Salvia* genus are typically herbaceous or shrubby perennials and are also popular as

Corresponding author
Uğur Tan, ugur.tan@adu.edu.tr

ornamental and garden plants (*Clebsch, 1997*; *Leontaritou et al., 2020*). However, the cultivation of this valuable plant, like many others, is threatened by salinity, a critical environmental challenge that severely limits plant growth and agricultural productivity (*Allakhverdiev et al., 2000*; *Atta et al., 2023*). Especially since *Salvia fruticosa* is primarily propagated through cuttings after being properly rooted under greenhouse conditions, the plant faces challenges when cultivated in field conditions characterized by saline soils or are irrigated with saline water. It struggles to adapt to the field with high salinity levels, which is an important stage of plants produced *via* cuttings, leading to reduced yield or even plant death. This causes significant transplanting shock, which can cause a setback in growth, especially under saline conditions (*Parida & Das, 2005*; *Krinis, Kasampalis & Siomos, 2023*). The primary issue is that excessive salt (NaCl) decreases the water potential in the soil, leading to osmotic stress, while the secondary issue is buildup of sodium ($Na^+$) and chloride ions ($Cl^-$) within plant cells as toxic. This ion accumulation interferes with nutrient absorption and causes significant cellular damage (*Hoang & Rehman, 2022*).

Several strategies have been studied to mitigate salinity stress in plants, including the use of fertilizers, nanoparticles, seed priming, plant hormones, and plant growth-promoting rhizobacteria (PGPR). Application of appropriate fertilizers can effectively alleviate the negative effects of salinity on plant growth (*Chen et al., 2009*). Nano-fertilizers such as nano-NPK (*Mustafa et al., 2022*) and calcium phosphate nanoparticles (*Nasrallah et al., 2022*) have shown promise in improving plant productivity and mitigating salinity stress. *Hajihashemi & Kazemi (2022)* highlighted the potential of nanoparticles to reduce salinity stress. Seed priming with phytohormones like GA3 and NaCl enhances seed germination and seedling establishment under salinity (*Iqbal et al., 2006*; *Sedghi, Nemati & Esmaielpour, 2010*; *Rhaman et al., 2020*). Halopriming also enhance vigor and metabolic reserves in seedlings (*Afzal, Rauf & Murtaza, 2008*; *Elouaer & Chérif, 2012*). PGPR has been effective in improving plant resilience under saline conditions (*Cappellari et al., 2023*). Exogenous hormone applications such as ethylene (*Riyazuddin et al., 2020*), salicylic acid (*Jamali, Eshghi & Kholdebarin, 2016*; *Sabzmeydani, Sedaghathoor & Hashemabadi, 2021*), gibberellin, and auxin (*Sá et al., 2020*) have been extensively studied for their role in improving plant tolerance to salinity by modulating plant defense mechanisms.

However, there are some concerns related to these applications; for instance, the use of nanoparticles presents the potential toxicity to plants. Moreover, the effectiveness and safety of nanoparticles can vary significantly across different plant species, making it carefully determine and optimize the dosages (*Ding et al., 2023*). Similarly, while seed priming treatments have shown efficacy in enhancing early plant growth, they often fail to produce significant improvements in later stages due to harsher environmental conditions in the field conditions (*Tan, 2024*). This challenge is also observed with plant growth-promoting rhizobacteria (PGPRs), where it performs well in laboratory settings often do not show the expected results in real-world agricultural environments (*Basu et al., 2021*). On the other hand, the effectiveness of hormones applied exogenously often depends on multiple treatments, which can decrease their usefulness due to the need for careful management of dosage and timing.

Salinity stress is often predictable, whether due to saline soils or high salt content in irrigation water. Therefore, preventive measures can be applied before stress occurs. Unlike other reactive approaches, the direct application of indole-3-butyric acid (IBA) can serve as a proactive measure to prepare plants in advance. It is also important identifying suitable plant species and effective treatments becomes important. However, *Salvia fruticosa*, a species with promising potential, has received significantly less attention compared to its close relative, *Salvia officinalis*. While both species belong to the same genus and share similar biochemical properties, *S. officinalis* is more widely distributed and commercially cultivated, making it a more common focus of research (*Boukhary et al., 2016*; *Tundis et al., 2017*). As a result, fewer studies have focused on *S. fruticosa* compared to other plants.

The aim of this study is to test the hypothesis: Applying IBA proactively to *Salvia fruticosa* cuttings before planting in high-salinity conditions will mitigate the negative effects of salinity on plant growth and development. This research also aims to fill a notable gap in the literature, as only a few studies have explored how *Salvia fruticosa* responds to salinity stress compared other plants.

## MATERIALS AND METHODS

### Preparation of plant material and application of IBA doses

In the experiment, cuttings were taken from seven year old *Salvia fruticosa* plants grown under field conditions on experimental field of Faculty of Agriculture at Aydin Adnan Menderes University during the vegetative growth stage in the autumn of 2023. Cuttings were collected from similar parts (upper fresh parts) of the plants free from disease or damage to ensure uniformity. Each cutting was thoroughly inspected, and any cutting that was uneven or damaged was discarded to ensure consistency.

For the preparation of IBA doses, Merck brand MW:203.24 IBA EC Number: 205-101-5 was used. The IBA was weighed with a precision balance and prepared as follows: 2–5 mL of solvent (ethanol) was used to dissolve the powder. Once completely dissolved, volume was adjusted with distilled water as 0 g/L: 500 ml distilled water, 1 g/L: 500 mg IBA/500 ml distilled water, and for 2 g/L: 1,000 mg IBA/500 ml distilled water. The cuttings were put in these IBA doses and waited for 15 s (quick dip) then planted in 24-cell plastic seedling trays filled with a 1:1 ratio of peat and perlite. The soil was watered before planting the cuttings, and the first watering of the plants was done 24 h later to prevent the IBA from being washed away. Watering was carried out once a week by giving 50 ml of water to each cell of tray during rooting period in the greenhouse conditions. After rooting, the seedlings were planted in pots where salinity conditions were applied to test the effectiveness of the IBA doses.

### Experimental design and salinity treatments

The experiment was designed as a two-factor randomized block design with four replications. The factors were arranged as three different IBA doses (0, 1, and 2 g/L) and four different salinity concentrations (0, 6, 12, and 18 dS m$^{-1}$). The salinity doses for *Salvia*

*fruticosa* were determined based on the doses used in *Kulak, Gul & Sekeroglu (2020)*, *Sheikhalipour et al. (2024)*, and *Taârit et al. (2012)*. A total of forty-eight (48) four-liter pots were used in the experiment, and the pots were filled equally with a 1:1 peat-perlite mixture to a volume of 3 liters. For the salinity treatments, Merck brand sodium chloride (NaCl) with EC number: 231-598-3 and MW: 58.44 was used. Saline water solutions were prepared by using an electrical conductivity (EC) WTW Cond 330i meter as 0, 6, 12, 18 dS $m^{-1}$, and watering was carried out with these solutions for each pot. Figure 1 shows the weekly accumulation of soil salinity, measured from the leachate after irrigation. According to the data, a logarithmic increase in salinity is observed.

The study was continued for 5 weeks, with salinity applications performed once a week with 600 ml of tap water. The water amount was adjusted to create at least 200 ml of runoff to accurately measure the salinity level in the leachate from the pots by using an EC meter.

## Measurements

*Salvia fruticosa* plants were measured and harvested 4 days after the final salinity application (Week 5). The following parameters were assessed: plant growth indicators, root parameters, root/shoot ratio (index), leaf chlorophyll content (SPAD), relative water content (RWC), relative membrane permeability (RMP), extraction yield (mg/g dw), IC50 (mg dw/mg DPPH), TEAC (mg/g dw), phenol (mg GAE/g dw), ABTS (mg TEAC/g dw), and flavonoid (mg rutin/g dw).

## Plant growth indicators

Plant heights (cm) for the plants in the experiment were recorded in centimeters from the ground level to the apex of the topmost leaf at the end of experiment (5th week). Flower spike length (cm) was measured with a ruler from the base to the tip of the flower spike in cm. Fresh shoot weights (g/plant) were harvested and immediately weighed to determine their initial mass. Then it was dried in the shade at room temperature to avoid losses in their chemical properties until a constant weight was achieved; later, dry shoot weights (g/plant) were measured. The drog (dried medicinal plant parts used for pharmacological purposes) yield (g/plant) was obtained from *Salvia fruticosa* by separating flowers and leaves from the stems and weighed. It is an important measurement for medicinal plants such as *Origanum*, *Salvia*, and *Mentha* as it directly relates to their yield as usable parts of plants for use as medicinal purposes.

## Root measurements

During harvest, roots were carefully excavated and cleaned to remove soil and debris without causing damage to roots. Root length (cm) was measured from the base to the tip using a ruler and recorded as cm. Fresh roots weight (g/plant) were immediately weighed using a precision scale to determine their fresh weight. The roots were then dried in an oven at 70 °C until a constant weight was achieved to obtain the dry root weight (g/plant).

## Root/shoot index (%)

The root/shoot index was calculated by dividing the dried root weight (DRW) by the dried shoot weight (DSW) and multiplying the result by 100. Root/shoot index = DRW/

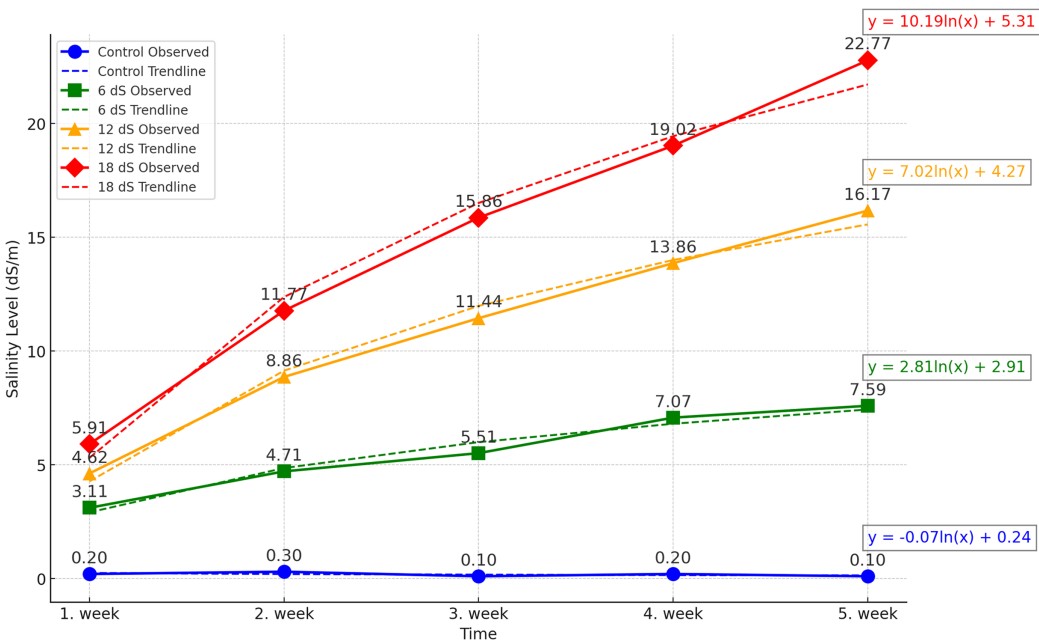

**Figure 1 Effect of different saline irrigations on soil salinity over time during experiment.**

DSW*100 (*Monk, 1966*). This index shows the allocation of biomass between the roots and shoots of the plant for understanding a plant's growth dynamics.

## Leaf chlorophyll content

Leaf chlorophyll content (SPAD) was measured using a "Konica Minolta SPAD-502 Plus" portable chlorophyll meter on fully developed leaves. The measurements were expressed as SPAD values following the method described by *Pask et al. (2012)*. For each plant, readings were taken from three leaves, with three measurements averaged per leaf. Measurements were taken of both upper part of plants (USPAD) and lower part of plants (LSPAD). All measurements were conducted between 11:00 a.m. and 2:00 p.m.

## Relative water content

The relative water content (RWC) was determined using the saturation drying method as described by *Tanentzap, Stempel & Ryser (2015)*. To ensure precision, each treatment was performed in triplicate. Leaves of *Salvia fruticosa* were harvested from each plant, and fresh weight (FW) was recorded immediately, and then the sample was submerged in deionized water for 24 h to achieve complete saturation. The saturated weight (TW) was afterwards recorded, and the leaves were dehydrated at 60 °C until a stable weight (DW) was attained. The RWC was computed utilizing the formula: RWC (%) = (FW − DW)/(TW − DW) × 100%.

## Extraction of samples and measuring extraction yield

For extraction, dried plant samples were ground in a grinder and then passed through a sieve to ensure homogeneity. From the ground powdered samples, 500 mg of dry sample

was used for extraction by mixing with 50 ml of 80% methanol in a shaking incubator at 40 °C for 2 h. Afterwards, 10 ml of the sample extracts were dried in an oven at 70 °C until a constant weight was achieved, and the extraction yield was calculated on a dry basis (*Maisuthisakul, Suttajit & Pongsawatmanit, 2007*). Sample extract immediately used after extraction.

## Relative membrane permeability

The third leaf of each plant that had fully developed was cut into two discs. These discs were then inserted in test tubes that contained twenty milliliters of distilled water. After vortexing the samples for 30 s, initial electrical conductivity (EC 0) of each sample was measured by the WTW Cond 330i meter for electrical conductivity (EC 0). The samples were then stored at 4 °C for 24 h, and the conductivity (EC 1) was measured again. The samples were then autoclaved for 15 min at 100 °C, then cooled to normal temperature, and the conductivity was measured for the third time (EC 2). The relative membrane permeability (RMP) was calculated using the following formula to assess the extent of cell membrane damage: RMP (%) = [(EC1−EC0)/(EC2−EC0)] × 100 (*Young, Wang & Leach, 1996*).

## DPPH assay

The antioxidant analysis has been conducted by using the DPPH test as described by studies (*Gadow, Joubert & Hansmann, 1997*; *Maisuthisakul, Suttajit & Pongsawatmanit, 2007*). A 100 μL extract was diluted to create four different concentrations and then added to 4 mL of freshly prepared DPPH (2,2-diphenyl-1-picrylhydrazyl radical) solution. The mixture shaken and incubated in the dark at room temperature for 30 min. The final solution's absorbance measured spectrophotometrically at 516 nm using a Thermo Fisher Scientific Multiskan GO microplate reader was used. microplate reader. The DPPH values of the samples are expressed in terms of trolox (TEAC) equivalents (mg/g dw) and as 50% inhibition IC50 (mg dw/mg DPPH) values.

## ABTS assay

The ABTS assay was performed following the methods described in a study (*Re et al., 1999*). To generate the ABTS radical cation (ABTS•+), a 7 mM ABTS solution was mixed with 2.45 mM potassium persulfate (1:1) and allowed to react in the dark for 16 h. The resulting ABTS•+ solution was then diluted with methanol or ethanol to reach an absorbance of 0.700. A 5 μL plant extract was added to 3.995 mL of the diluted ABTS•+ solution and incubated for 30 min. The absorbance of ABTS•+ was measured and expressed in terms of trolox equivalents (mg TEAC/g dw).

## Flavonoid content

For flavonoid content determination, 0.5 mL of the extract was mixed with 2.5 mL of distilled water and 150 μL of 5% $NaNO_2$, then vortexed. After 5 min, 300 μL of 10% AlCl3 was added, and the mixture was allowed to react for another 5 min. Subsequently, 1 mL of 1 M NaOH was added, and the final volume was adjusted to 5 mL with distilled water. The absorbance was measured at 510 nm using a Thermo Fisher Scientific Multiskan GO

microplate reader, and the results were expressed in terms of Rutin trihydrate (MW: 664.56) equivalents (mg rutin/g dw) (*Chang et al., 2006*).

## Phenolic content

Phenolic content analysis was conducted following the method described in previous studies (*Skerget et al., 2005*). A 0.5 mL extract was mixed with 2.5 mL of 0.1 N Folin-Ciocalteu phenol reagent and 2 mL of $Na_2CO_3$ (75 g/L). The mixture was incubated at 50 °C for 5 min and then immediately cooled. The absorbance of the samples was measured at 760 nm using a Thermo Fisher Scientific Multiskan GO microplate reader, and the results were expressed in terms of gallic acid equivalents (mg GAE/g dw).

## Statistical analyses

The data were subjected to an analysis of variance (one-way ANOVA) followed by comparison of multiple treatment levels with controls using Fisher's *post hoc* LSD (least significant difference) test. All data were analyzed using JMP Pro software (SAS Institute, Cary, NC, USA). Also, for visualization, JMP Pro and Python's matplotlib library were used.

## RESULTS

The ANOVA (Table 1) and bar charts (Fig. 2) show the effects of different salinity (S) levels and IBA applications on plant height (cm) (PH), root length (cm) (RL), fresh root weight (g/plant) (FRW), fresh shoot weight (g/plant) (FSW), dried root weight (g/plant) (DRW), dried shoot weight (g/plant) (DSW), root/shoot weight index (%) (DRW/DSW), and drog (g/plant).

As salinity increases, plant height decreases significantly. At 0 dS/m (no salinity), the highest plant height is recorded at 54 cm, but as salinity reaches 18 dS/m, plant height drops to 27.75 cm. IBA application positively influences plant height across all salinity levels. Without IBA (0 g/L), the lowest plant height is observed at 27.34 cm. However, at 2 g/L of IBA, plant height increases substantially to 55.75 cm. The salinity*IBA interaction shows the highest plant height (73.38 cm) occurs at 0 dS/m and 2 g/L IBA. The flower spike length (FSL) shows the increasing salinity levels negatively impact flower spike growth. At 0 dS/m (no salinity), the FSL is highest at 29.5 cm, but it decreases progressively with higher salinity, reaching just 13.06 cm at 18 dS/m. IBA application positively influences FSL. Without IBA (0 g/L), the FSL is relatively low at 16.61 cm. However, with 2 ppm IBA, FSL increases significantly to 26.48 cm. The combination of salinity with IBA, the highest FSL (34 cm), occurs at 0 dS/m with 2 g/L IBA. The longest RL (root length) is 30.5 cm, was obtained at 0 dS/m (no salinity), however, as salinity increases to 6 dS/m, RL decreases to 27.33 cm, and it drops further to 19.5 cm at 12 dS/m and 15.64 cm at 18 dS/m. It shows that high salinity severely prevents root development. IBA (0 g/L), RL value was found as 21.94 cm; application of 1 g/L IBA increases RL to 24.19 cm, and at 2 g/L IBA, RL is slightly decreased to 23.6 cm with no statistical difference ($p > 0.05$). When examining the IBA and salinity interactions, while high

**Table 1 ANOVA and effects of salinity and IBA on plant growth parameters.**

| Aplications | | PH | FSL | RL | FRW | FSW | DRW | DSW | R/S | DROG |
|---|---|---|---|---|---|---|---|---|---|---|
| | | *Salinity (S)* | | | | | | | | |
| 0 dS/m | | 54.00 a | 29.50 a | 30.50 a | 11.37 a | 21.69 a | 1.44 a | 4.94 a | 28.51 ab | 2.11 a |
| 6 dS/m | | 47.54 b | 24.75 b | 27.33 a | 10.49 a | 21.25 a | 1.24 a | 4.16 b | 30.22 ab | 2.12 a |
| 12 dS/m | | 30.08 c | 14.31 c | 19.50 b | 5.58 b | 8.66 b | 0.65 b | 2.50 c | 24.08 b | 1.19 b |
| 18 dS/m | | 27.75 c | 13.06 c | 15.64 b | 4.94 b | 7.32 b | 0.60 b | 1.76 d | 35.73 a | 0.95 b |
| | | *IBA (I)* | | | | | | | | |
| 0 g/L | | 27.34 c | 16.61 b | 21.94 | 5.83 b | 11.77 b | 0.76 b | 2.75 b | 25.78 b | 1.29 b |
| 1,000 g/L | | 36.44 b | 18.13 b | 24.19 | 6.33 b | 14.25 b | 0.78 b | 3.03 b | 28.23 b | 1.47 b |
| 2,000 g/L | | 55.75 a | 26.48 a | 23.60 | 12.12 a | 18.17 a | 1.40 a | 4.24 a | 34.90 a | 2.02 a |
| | | *IxS* | | | | | | | | |
| 0 dS/m | 0 g/L | 43.38 cd | 28.75 ab | 35.50 a | 10.79 bc | 22.49 ab | 1.54 ab | 4.99 ab | 30.45 | 1.99 a |
| | 1 g/L | 45.25 cd | 25.75 bc | 28.25 b | 6.60 de | 21.97 ab | 0.86 cd | 4.71 abc | 18.50 | 2.09 a |
| | 2 g/L | 73.38 a | 34.00 a | 27.75 bc | 16.72 a | 20.62 abc | 1.91 a | 5.13 a | 36.58 | 2.26 a |
| 6 dS/m | 0 g/L | 46.50 bcd | 24.50 bc | 26.50 bcd | 9.20 cde | 20.49 abc | 1.05 cd | 3.94 bc | 27.57 | 2.27 a |
| | 1 g/L | 39.00 de | 23.00 bc | 27.00 bcd | 9.55 bcd | 19.02 bc | 1.18 bc | 3.77 c | 31.45 | 1.79 ab |
| | 2 g/L | 57.13 b | 26.75 ab | 28.50 ab | 12.72 b | 24.24 a | 1.50 b | 4.78 abc | 31.64 | 2.31 a |
| 12 dS/m | 0 g/L | 6.00 f | 1.94 e | 14.00 ef | 1.03 g | 1.55 f | 0.16 f | 0.82 e | 21.73 | 0.34 e |
| | 1 g/L | 30.75 e | 10.75 d | 21.00 cde | 3.18 fg | 8.12 e | 0.41 ef | 1.87 de | 22.03 | 1.09 cd |
| | 2 g/L | 53.50 bc | 26.50 abc | 23.50 bcd | 10.60 bc | 16.32 cd | 1.24 bc | 4.80 abc | 28.49 | 2.13 a |
| 18 dS/m | 0 g/L | 13.50 f | 11.25 d | 11.75 f | 2.31 g | 2.55 f | 0.30 ef | 1.27 de | 23.35 | 0.55 de |
| | 1 g/L | 30.75 e | 13.00 d | 20.50 de | 5.99 ef | 7.89 e | 0.68 de | 1.77 de | 40.95 | 0.91 cd |
| | 2 g/L | 39.00 de | 18.67 cd | 14.67 ef | 8.45 cde | 11.51 de | 0.96 cd | 2.24 d | 42.89 | 1.38 bc |
| Anova | df | *P value* | | | | | | | | |
| S | 3 | <0.0001** | <0.0001** | <0.0001** | <0.0001** | <0.0001** | <0.0001** | <0.0001** | 0.026* | <0.0001** |
| I | 3 | <0.0001** | <0.0001** | 0.4279ns | <0.0001** | <0.0001** | <0.0001** | <0.0001** | 0.0188* | <0.0001** |
| IxS | 2 | 0.0002** | 0.0099** | 0.0157* | 0.0052** | 0.0019** | 0.0037** | 0.0011** | 0.0677ns | 0.0021** |

**Note:**
Plant height (PH), fresh shoot length (FRL), root length (RL), and fresh root weight (FRW), fresh shoot weight (FSW), Dried root weight (DRW), dried shoot weight (DSW), root/shoot index (%), Drog (dry plant separated from stems), lowercase letters indicate statistically significant groupings. *$p < 0.05$, **$p < 0.01$.

salinity reduces root length, IBA application, particularly at higher concentrations like 2 g/L, helps mitigate this effect.

The fresh root weight (FRW) has the highest value with 11.37 g/plant at 0 dS/m (no salinity); however, as salinity increases to 6 dS/m, FRW slightly decreases to 10.49 g/plant, and at 12 dS/m, it drops significantly to 5.58 g/plant. The lowest FRW is observed at 18 dS/m, with 4.94 g/plant. IBA application positively affects FRW; without IBA (0 g/L), FRW has a low value of 5.83 g/plant; however, with 1 g/L IBA, FRW increases to 6.33 g/plant, and at 2 g/L IBA, it significantly rises to 12.12 g/plant. The 2 g/L IBA at 0 dS/m has highest FRW (16.72 g/plant) value. FSW value has the highest value at 0 dS/m (no salinity) with 21.69 g/plant, but as salinity increases, FSW decreases slightly to 21.25 g/plant at 6 dS/m and significantly drops to 8.66 g/plant at 12 dS/m and 7.32 g at 18 dS/m. The dry root weight (DRW) was negatively affected by salinity, with higher salinity levels leading to significant reductions. At 0 dS/m (no salinity), the highest DRW is 1.44 g/plant; as salinity

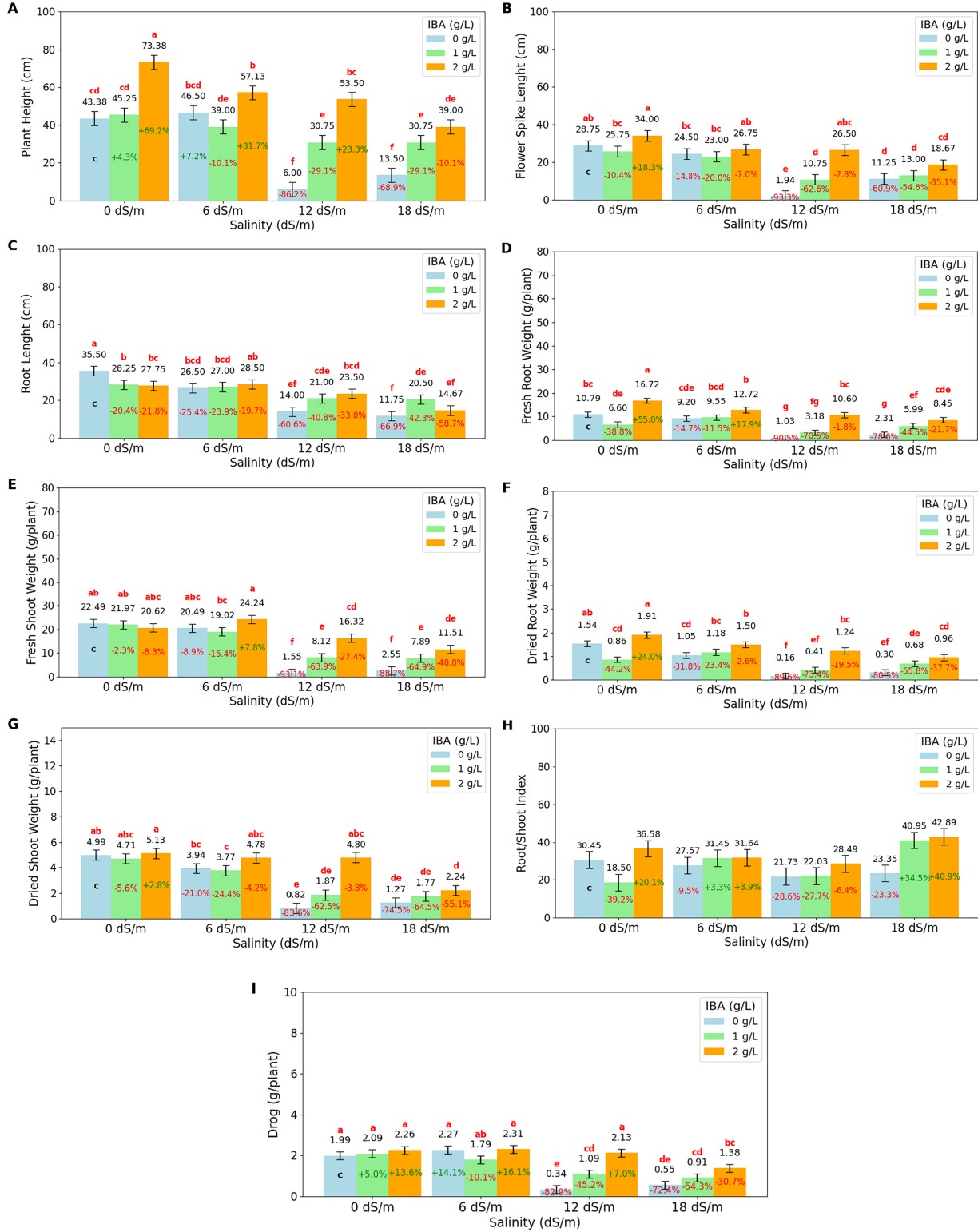

**Figure 2 Bar charts of growth parameters affected by salinity and IBA dosages.** (A) Plant height, (B) flower spike length, (C) root length, (D) fresh root weight, (E) fresh shoot weight, (F) dried root weight, (G) dried shoot weight, (H) root/shoot Index, (I) drog. The charts show the magnitude of the respective measurement parameter, providing a visual representation of the data trends. The percentage values inside of bars indicate the extent of the change compared to the control.

increases, DRW slightly decreases to 1.24 g/plant at 6 dS/m, and then significantly drops to 0.65 g/plant at 12 dS/m and 0.6 g/plant at 18 dS/m. IBA application positively affects DRW, as IBA (0 g/L) is 0.76 g/plant, with 1 g/L IBA DRW increases to 0.78 g/plant, and at 2 g/L IBA, it increases significantly to 1.40 g/plant. At 0 dS/m (no salinity), the highest DSW is 4.94 g/plant; DSW decreases to 1.76 g/plant at 18 dS/m. The root/shoot index is positively affected by IBA application at 2 g/L, increasing to 34.9%. At higher salinity levels (12 and 18 dS/m), the ratio drops to 24.08% at 12 dS/m but rises abruptly to 35.73% at 18 dS/m, indicating a significant increase in root allocation as a survival response under severe stress. The drog value, representing the plant's dried weight after separation from its stems, is a vital measurement for medicinal plants as it directly relates to their yield and potential for producing valuable medicinal compounds. Under non-saline conditions (0 dS/m), the highest drog value of 2.11 g/plant has maximum yield potential; at moderate salinity (6 dS/m), the drog value remains stable at 2.12 g/plant, showing that this level of stress does not significantly impact overall yield. However, as salinity increases, the drog value drops significantly, to 1.19 g/plant at 12 dS/m and to 0.95 g/plant at 18 dS/m, indicating a substantial reduction in yield. IBA application, particularly at higher concentrations, mitigates some of these negative effects on yield.

The ANOVA (Table 2) and bar charts (Fig. 3) show the effects of different salinity (S) levels and IBA applications on RWC, RMP, L-SPAD, and U-SPAD.

RWC is an important indicator of a plant's ability to retain water, especially under stress conditions like salinity. The value shows that increasing salinity levels negatively impact RWC, with the highest value observed at 0 dS/m (90.03%), showing optimal water retention in the absence of salinity stress. As salinity increases to 6 dS/m, RWC drops to 84.24%; this trend continues at 12 dS/m, where RWC further decreases to 74.99% and reaches its lowest at 18 dS/m (67.34%). The application of IBA positively influences RWC. As IBA (0 g/L), RWC has relatively low value at 66.62%; however, with 1 g/L IBA, RWC increases to 82.04% and reaches 88.79% with 2 g/L IBA under stress. The interaction between salinity and IBA (IxS), shows that IBA mitigates the negative effects of salinity on RWC. At 6 dS/m, 2 g/L IBA maintains a high RWC of 89.17%, and even at 18 dS/m, RWC remains relatively high at 86.46% with 2 g/L IBA. Relative Membrane Permeability (RMP) is also an important value that shows cell membrane integrity in plants, with higher RMP values indicating increased membrane damage and stress. The data shows that salinity significantly impacts RMP, with the lowest value (8.26%) observed at 0 dS/m, reflecting minimal stress and healthy cell membranes under non-saline conditions. As salinity increases to 6 dS/m, RMP rises to 30.46%; it continues to rise at 12 dS/m, where RMP reaches 43.57%, and peaks at 18 dS/m with a value of 61.50%. The application of IBA shows a protective effect on membrane integrity. IBA (0 g/L), RMP is relatively high at 42.76%, indicating significant membrane damage. However, with 1 g/L IBA, RMP decreases to 32.88%, and further to 32.20% with 2 g/L IBA, that higher concentrations of IBA effectively reduce membrane permeability and protect cell membranes from stress-induced damage. When considering the interaction between salinity and IBA (IxS), the values show that IBA mitigates the negative effects of salinity on RMP. At 6 dS/m, 2 g/L

**Table 2 ANOVA and effects of salinity and IBA on RWC (%), RMP (%) and SPAD values.**

| Aplications | | RWC(%) | RMP(%) | LSPAD | USPAD |
|---|---|---|---|---|---|
| | | *Salinity (S)* | | | |
| 0 dS/m | | 90.03 a | 8.26 d | 35.47 a | 38.32 a |
| 6 dS/m | | 84.24 b | 30.46 c | 37.11 a | 40.42 a |
| 12 dS/m | | 74.99 c | 43.57 b | 30.73 b | 29.94 b |
| 18 dS/m | | 67.34 d | 61.50 a | 26.96 c | 29.63 b |
| | | *IBA (I)* | | | |
| 0 g/L | | 66.62 c | 42.76 a | 31.49 | 32.16 c |
| 1,000 g/L | | 82.04 b | 32.88 b | 33.39 | 34.54 b |
| 2,000 g/L | | 88.79 a | 32.20 b | 32.83 | 37.03 a |
| | | *IxS* | | | |
| 0 dS/m | 0 g/L | 90.46 a | 11.04 h | 37.28 ab | 39.78 ab |
| | 1 g/L | 89.05 a | 6.04 i | 36.00 abc | 39.08 abc |
| | 2 g/L | 90.59 a | 7.71 hi | 33.13 bcd | 36.10 bc |
| 6 dS/m | 0 g/L | 76.62 b | 32.30 f | 39.17 a | 41.23 a |
| | 1 g/L | 86.94 a | 30.84 fg | 37.18 ab | 39.15 abc |
| | 2 g/L | 89.17 a | 28.25 g | 34.98 abc | 40.88 a |
| 12 dS/m | 0 g/L | 58.33 c | 56.37 c | 25.92 ef | 23.45 e |
| | 1 g/L | 77.67 b | 33.90 f | 32.53 cd | 31.08 d |
| | 2 g/L | 88.95 a | 40.43 e | 33.75 bcd | 35.30 c |
| 18 dS/m | 0 g/L | 41.08 d | 71.32 a | 23.59 f | 24.20 e |
| | 1 g/L | 74.50 b | 60.75 b | 27.85 ef | 28.85 d |
| | 2 g/L | 86.46 a | 52.43 d | 29.45 de | 35.85 bc |
| Anova | df | *P value* | | | |
| S | 3 | <0.0001** | <0.0001** | <0.0001** | <0.0001** |
| I | 3 | <0.0001** | <0.0001** | 0.2428ns | <0.0002** |
| IxS | 2 | <0.0001** | <0.0001** | 0.0033** | <0.0001** |

**Note:**
Relative membrane content (RMC), relative membrane permeability (RMP), LSPAD (lower leaves), USPAD (upper leaves). Lowercase letters indicate statistically significant groupings. $^*p < 0.05$, $^{**}p < 0.01$.

IBA maintains a lower RMP (28.25%), and even under severe salinity (18 dS/m), RMP remains relatively lower (52.43%) with IBA treatment.

LSPAD and USPAD are the indicators of chlorophyll content in the lower and upper parts of leaves, respectively, providing insight into the plant's photosynthetic capacity and response to stress, such as salinity. The lower parts of leaves (LSPAD) are typically more vulnerable and are often the first to show signs of stress, while the upper parts (USPAD) might be less immediately affected, making the comparison between these two parameters important to understanding how stress impacts the plant. Under non-saline conditions (0 dS/m), both LSPAD and USPAD values are high, with USPAD (38.32) slightly higher than LSPAD (35.47); when salinity increases to 6 dS/m, both LSPAD and USPAD show a slight increase. However, as salinity stress intensifies at level (18 dS/m), LSPAD drops to 26.96, and USPAD to 29.63. The application of IBA shows a protective effect on chlorophyll content, particularly in the upper leaves. Without IBA (0 g/L), both LSPAD

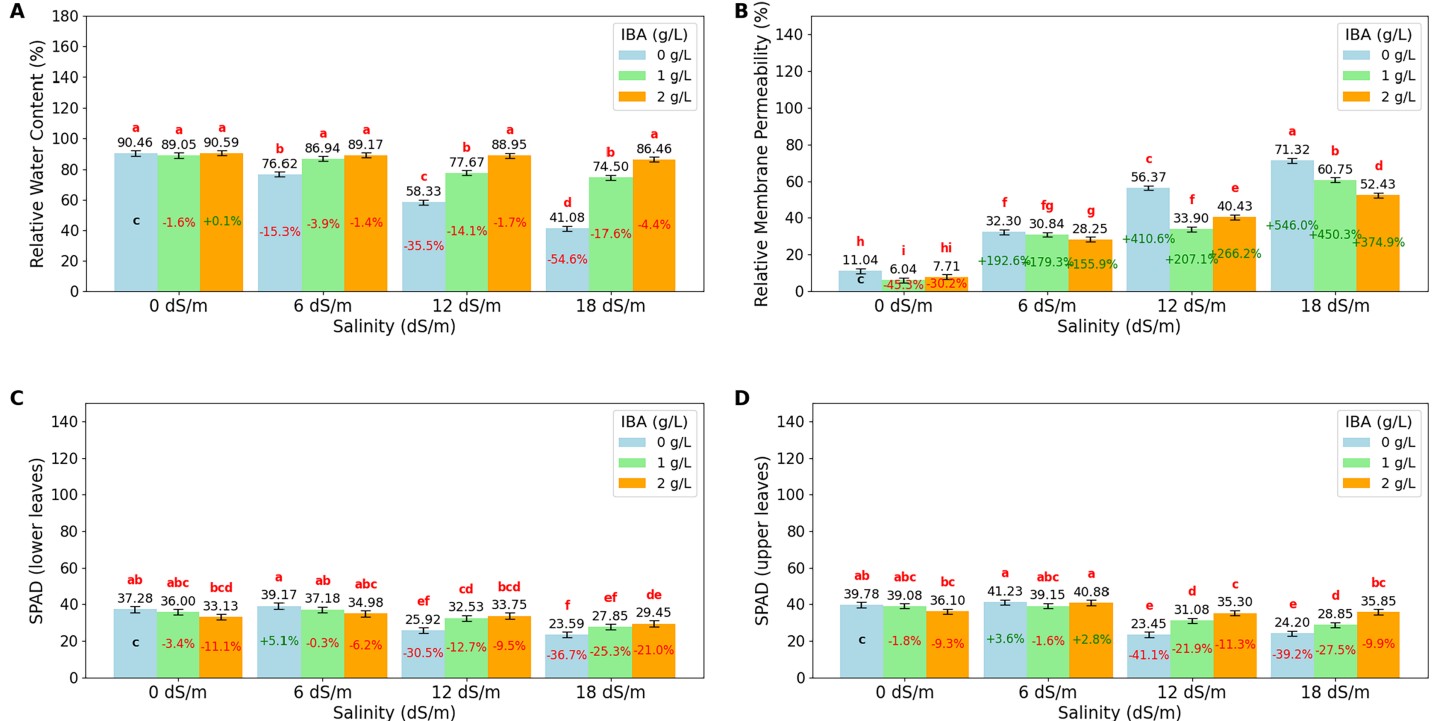

**Figure 3 Bar charts of physiological parameters affected by salinity and IBA dosages.** (A) Relative membrane content, (B) relative membrane permeability, (C) SPAD (lower leaves), (D) SPAD (upper leaves). The charts show the magnitude of the respective measurement parameter, providing a visual representation of the data trends. The percentage values inside of bars indicate the extent of the change compared to the control.

and USPAD have lower values, however, with increasing concentrations of IBA, especially at 2 g/L, both LSPAD and USPAD values improve. The interaction effects of salinity and IBA, show that IBA helps mitigate the negative impact of salinity on chlorophyll content. Even at higher salinity levels (12 and 18 dS/m), IBA application maintains higher LSPAD and USPAD values. Also, it is noticeable that LSPAD was affected more than USPAD values as salinity stress increases.

The ANOVA (Table 3) and bar charts (Fig. 4) show the effects of different salinity (S) levels and IBA applications on extract yield (mg/g dry weight) (EY), IC50 (mg dw/mg DPPH), trolox equivalent antioxidant capacity (TEAC mg/g dw), phenol content (mg GAE/g dw) (gallic acid equivalent), ABTS (mg TEAC/g dw) (trolox equivalent) and flavonoid (mg rutin/g dw) (mg rutin trihydrate equivalent/g dry weight).

Extraction yield (EY) measures the efficiency of extracting bioactive compounds from plant material. The results show that salinity stress increases EY, with the highest yield observed at 18 dS/m (510.08 mg/g dw). The application of IBA increases the EY, with control IBA (0 g/L), EY was found to be 401.13 mg/g dw, and with 1 g/L IBA, EY increases to 507.13 mg/g dw, and at 2 g/L IBA, EY remains as 502.13 mg/g dw, slightly lower than at 1 g/L. When examined, the I*S interactions, the highest EY values are achieved, particularly under severe stress conditions with 18 dS/m with 1 g/L IBA, EY reaches a value of 610.50 mg/g dw.

**Table 3 ANOVA on the effects of salinity and IBA on EY, IC50, TEAC, phenol, ABTS, and flavonoid contents.**

| Aplications | | EY (mg/g DW) | IC50 (mg dw/mg DPPH) | TEAC (mg/g dw) | Phenol (mg GAE/g dw) | ABTS (mg TAEC/g dw) | Flavonoid (mg rutin/g dw) |
|---|---|---|---|---|---|---|---|
| *Salinity (S)* | | | | | | | |
| 0 dS/m | | 442.42 d | 2.29 d | 92.44 a | 83.84 a | 309.75 a | 309.07 a |
| 6 dS/m | | 457.83 c | 2.39 c | 91.61 a | 57.86 b | 242.08 b | 198.13 b |
| 12 dS/m | | 470.17 b | 2.60 b | 88.52 b | 33.33 c | 194.67 c | 119.73 c |
| 18 dS/m | | 510.08 a | 2.86 a | 80.26 c | 25.53 d | 174.25 d | 82.53 d |
| *IBA (I)* | | | | | | | |
| 0 ppm | | 401.13 b | 2.94 a | 78.13 c | 44.96 c | 215.65 b | 144.80 c |
| 1,000 ppm | | 507.13 a | 2.40 b | 90.97 b | 50.64 b | 204.52 c | 177.40 b |
| 2,000 ppm | | 502.13 a | 2.25 c | 95.54 a | 54.82 a | 270.40 a | 209.90 a |
| *IxS* | | | | | | | |
| 0 dS/m | 0 ppm | 436.75 f | 2.26 ef | 96.35 bc | 89.65 a | 307.83 a | 308.80 b |
| | 1,000 ppm | 412.25 g | 2.27 def | 90.39 d | 82.78 b | 303.58 a | 280.80 c |
| | 2,000 ppm | 478.25 e | 2.34 d | 90.59 d | 79.09 c | 317.83 a | 337.60 a |
| 6 dS/m | 0 ppm | 369.25 i | 2.75 c | 81.64 e | 54.91 e | 259.83 c | 168.00 e |
| | 1,000 ppm | 515.50 c | 2.19 f | 98.47 ab | 59.62 d | 214.08 d | 208.40 d |
| | 2,000 ppm | 488.75 de | 2.21 ef | 94.72 c | 59.07 d | 252.33 c | 218.00 d |
| 12 dS/m | 0 ppm | 392.75 h | 3.37 a | 68.29 g | 15.38 j | 132.83 f | 45.60 i |
| | 1,000 ppm | 490.25 d | 2.24 ef | 97.49 abc | 39.64 g | 168.83 e | 146.40 f |
| | 2,000 ppm | 527.50 b | 2.19 f | 99.78 a | 44.98 f | 282.33 b | 167.20 e |
| 18 dS/m | 0 ppm | 405.75 g | 3.39 a | 66.22 g | 19.91 i | 162.08 e | 56.80 i |
| | 1,000 ppm | 610.50 a | 2.91 b | 77.51 f | 20.51 i | 131.58 f | 74.00 h |
| | 2,000 ppm | 514.00 c | 2.27 de | 97.06 abc | 36.16 h | 229.08 d | 116.80 g |
| Anova | df | *P value* | | | | | |
| S | 3 | <0.0001** | <0.0001** | <0.0001** | <0.0001** | <0.0001** | <0.0001** |
| I | 3 | <0.0001** | <0.0001** | <0.0001** | <0.0001** | <0.0001** | <0.0001** |
| IxS | 2 | <0.0001** | <0.0001** | <0.0001** | <0.0001** | <0.0001** | <0.0001** |

**Note:**
Extraction yield (EY), TAEC (DPPH), IC%50 (DPPH), Lowercase letters indicate statistically significant groupings. *p < 0.05, **p < 0.01.

DPPH (IC50) and DPPH (TEAC) are both measures of antioxidant activity in plants, but they provide different insights. DPPH (IC50) measures the concentration required to inhibit 50% of the DPPH radical, with lower values indicating higher antioxidant activity. Conversely, DPPH (TEAC) expresses antioxidant capacity in terms of Trolox equivalent antioxidant capacity, with higher values indicating greater antioxidant power. As salinity levels increase, DPPH (IC50) values rise, indicating a decline in the plant's ability to neutralize free radicals. Starting with an IC50 of 2.29 mg dw/mg DPPH at 0 dS/m (no salinity), it increases to 2.86 mg dw/mg DPPH at 18 dS/m, showing that severe salinity stress reduces antioxidant activity. Also, DPPH (TEAC) values decrease as salinity increases, from 92.44 TEAC mg/g dw at 0 dS/m to 80.26 TEAC mg/g dw at 18 dS/m, reflecting a reduction in the overall antioxidant capacity of the plant under stress. The application of IBA influences both DPPH (IC50) and DPPH (TEAC). IC50 is relatively

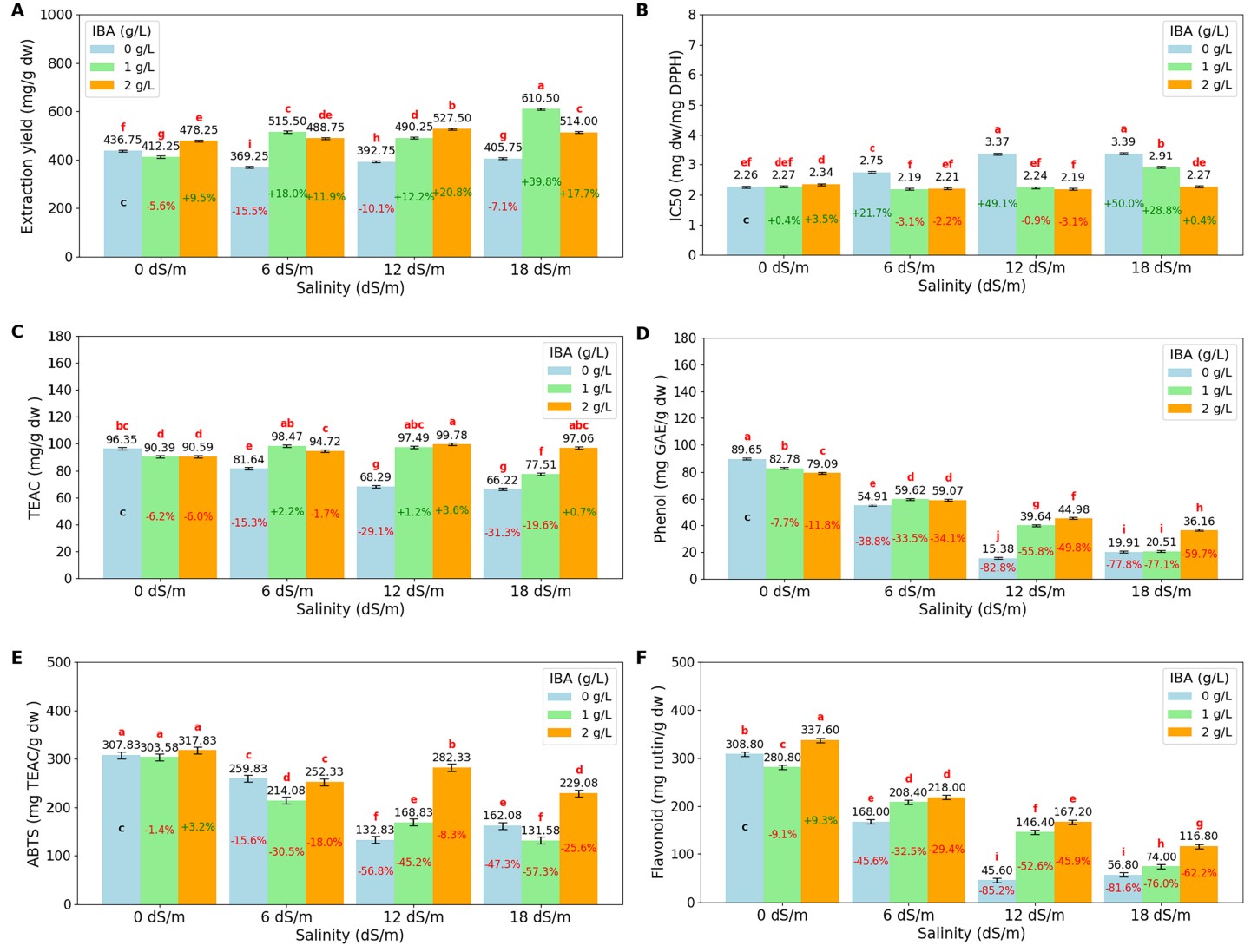

**Figure 4 Bar charts of antioxidants parameters affected by salinity and IBA dosages.** (A) Extraction Yield, (B) IC50 (DPPH), (C) TEAC (DPPH), (D) phenol content, (E) ABTS, (F) flavonoid content. The charts show the magnitude of the respective measurement parameter, providing a visual representation of the data trends. The percentage values inside of bars indicate the extent of the change compared to the control.

high with 2.94 mg dw/mg DPPH, IBA (0 g/L), which means lower antioxidant activity, while TEAC has lower value with 78.13 TEAC mg/g dw. With the application of 1 and 2 g/L IBA, IC50 decreases to 2.40 and 2.25, respectively. Simultaneously, TEAC increases to 90.97 and 95.54, respectively. IBA and salinity interactions show that at 18 dS/m with 2 g/L IBA, IC50 is reduced to 2.27 mg dw/mg DPPH, and TEAC is increased with 97.06 TEAC mg/g dw, showing that IBA effectively boosts the plant's antioxidant defenses under severe stress.

Phenol content shows that increasing salinity significantly reduces phenol content. Phenol content was found to be 83.84 mg GAE/g dw at 0 dS/m, as salinity increases to 6 dS/m, phenol content drops to 57.86 mg GAE/g dw, and further declines to 33.33 mg/g at
12 dS/m and 25.53 mg GAE/g dw at 18 dS/m. The application of IBA helps counteract this effect. Phenol content is found relatively low with 44.96 mg GAE/g dw at 0 g/L IBA, however, with 1 g/L IBA, phenol content increases to 50.64 mg GAE/g dw, and with 2 g/L IBA, it has a value of 54.82 mg/g. When salinity*IBA interactions were examined, the data showed that IBA helps mitigate the reduction in phenol content. The high salinity effects (18 dS/m), phenol content with 2 g/L IBA has 36.16 mg GAE/g dw, higher than 0 g/L IBA application. The ABTS assay measures the overall antioxidant potential by evaluating the plant's ability to scavenge free radicals. Under non-stress conditions (0 dS/m), the ABTS value is high with 309.75 mg TEAC/g dw, indicating strong antioxidant activity. However, as salinity increase, the ABTS value decreases significantly, dropping to 174.25 mg TEAC/ g dw at 18 dS/m. Also, flavonoid content shows similar decreases like ABTS as salinity levels increases. At 0 dS/m, the flavonoid content is high at 309.07 mg rutin/g dw, and it decreases to 82.53 mg rutin/g dw at 18 dS/m. Both values are increase slightly with the application of IBA doses under salinity conditions.

## Principal component analysis

Principal component analysis (PCA) is a statistical technique used to analyze and simplify large and complex datasets. In this study, PCA was used to explore the patterns among measured parameters that were affected by different salinity conditions and evaluate relation between parameters.

For *Salvia fruticosa* grown under control condition (0 dS/m NaCl), the total variation was divided into 11 principal components (PCs) (Fig. 5A). Among these, the first five PCs had eigenvalues greater than 1, this shows their significance in explaining the relation among the parameters under control condition (0 dS/m). These five PCs together accounted for 86.256% of the total variability in the measured parameters showing a strong association with the traits under control conditions. In contrast, the remaining PCs, with lower eigenvalues, collectively contributed only 13.744% to the overall variability. Further biplot (Fig. 5A) was used to analyze the correlations. Under control conditions (0 dS/m), all parameters were distributed within the correlation ellipse formed by the first two principal components (Component-1 and Component-2), together explained 58.5% of the total variation. Several trait correlations were observed: DRW, FSL, FRW, root/shoot index (DRW/DSW), drog, and Flav. showed positive correlations with each other; RL, TEAC, and Phn. were positively correlated, and LSPAD and USPAD also showed a positive correlation with each other. Conversely, LSPAD and USPAD exhibited negative correlations with RWC, IC50, and PH. The vector lengths originating from the center of the biplot depicted the strength of correlations among the measured parameters. Traits such as DRW, FRW, and PH displayed long vectors, show higher variation, whereas RWC and DROG exhibited the least variability.

For *Salvia fruticosa* grown under 6 dS/m salinity stress, the total variation was divided into 11 principal components (PCs). The first five PCs had eigenvalues greater than 1 (Fig. 5B). The cumulative contribution of these five PCs accounted for 87.722% of the measured parameters, showing a strong association with the traits under salinity stress. In contrast, the remaining PCs, with lower eigenvalues, collectively contributed 12.278% to

the overall variability. Under 6 dS/m salinity conditions, all parameters were distributed within the correlation ellipse formed by the first two principal components (Component-1 and Component-2), which together explained 63.3% of the total variation. Several correlations among traits were observed: DSW, FSW, PH, and FSL showed positive correlations with each other, while they were negatively correlated with LSPAD. RL, Phn., EV, RWC, and R/S index were positively correlated, but negatively correlated with IC50, RMP, ABTS, and USPAD. LSPAD also displayed a negative correlation with FRW and DRW. Traits such as EV, Flav., FSW, and DSW displayed long vectors, meaning higher variation, while RL, RMP, and USPAD exhibited the low variability.

Under 12 dS/m salinity stress, the total variation was divided into 11 principal components (PCs) (Fig. 5C). The first three PCs had eigenvalues greater than 1. The cumulative contribution of the first two PCs accounted for 84.251% of the total variability in the measured parameters under 12 dS/m conditions. Unlike other salinity levels, the first principal component (PC-1) alone contributed 75.237% of the total variability, showing its importance for these traits under 12 dS/m stress. In contrast, the remaining PCs, with lower eigenvalues, collectively contributed 15.749% to the overall variability. Under 12 dS/m conditions, all parameters were distributed within the correlation ellipse formed by the first two principal components (Component-1 and Component-2), which together explained 84.21% of the total variation. The results showed that RMP and IC50 were positively correlated with each other but negatively correlated with all other parameters. The vector lengths originating from the center of the biplot, validating the correlations under 12 dS/m conditions. In terms of variability, the traits R/S, RL, FSW, and LSPAD exhibited the least variability.

Under 18 dS/m salinity stress, the total variation was divided into 11 principal components (PCs) (Fig. 5D). The first three PCs had eigenvalues greater than 1. The cumulative contribution of these five PCs accounted for 85.762% of the total variability in the measured parameters, under 18 dS/m conditions. In contrast, the remaining PCs, with lower eigenvalues, collectively contributed 14.238% to the overall variability. The biplot was used to analyze trait correlations and their magnitudes of variation. Under 18 dS/m conditions, all parameters were distributed within the correlation ellipse formed by the first two principal components (Component-1 and Component-2), which together explained 79.7% of the total variation. The following trait correlations were observed: IC50 and RMP showed a positive correlation with each other but were negatively correlated with DSW, RWC, PH, and FRW. Additionally, DROG and USPAD exhibited a strong positive correlation with each other, as did FSL and Flav. In terms of variability, the traits IC50, Phn, Flav., and USPAD displayed long vectors, indicating higher variation, while DSW, FSW, and DRW/DSW exhibited the low variability.

In order to evaluate relation of parameters and treatments in this study, two-way cluster heatmap were used (Fig. 6). The two-way hierarchical clustering heatmap shows how different IBA and salinity (NaCl) treatments influence various growth, physiological, and biochemical parameters. By examining the color patterns and clustering of treatment combinations and parameters, it can be better understood how these factors interact and impact plant responses. According to results treatments with high salinity (18 dS/m) and

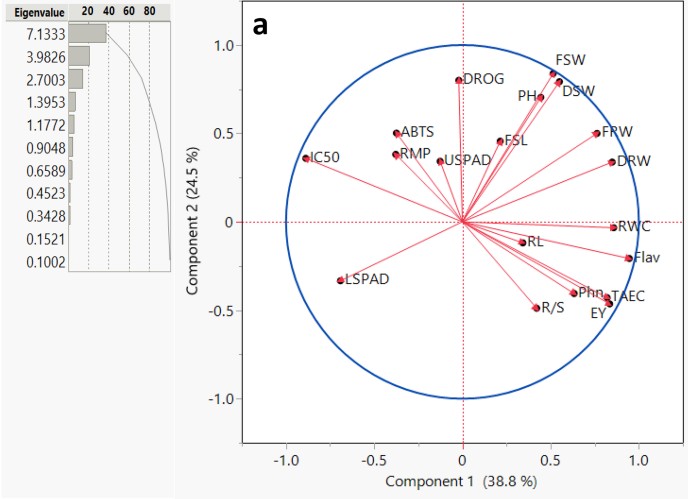
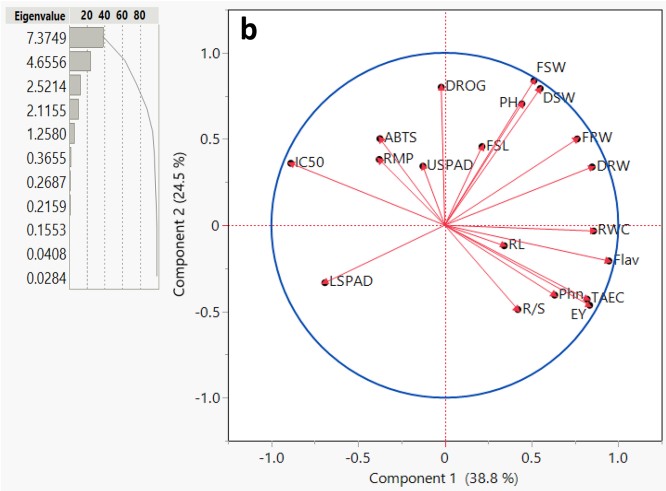
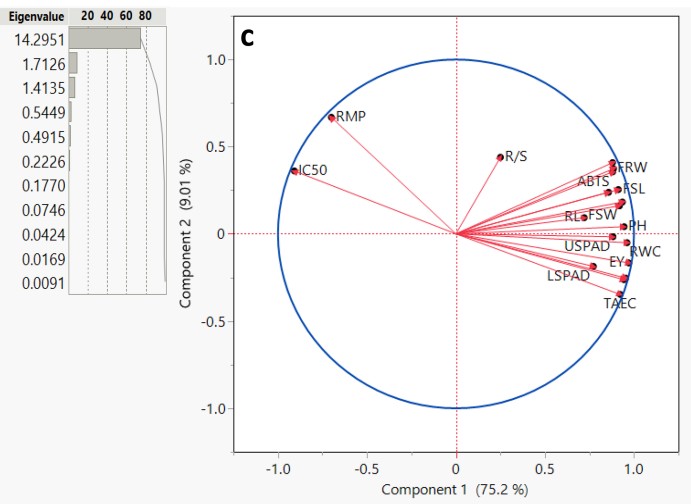
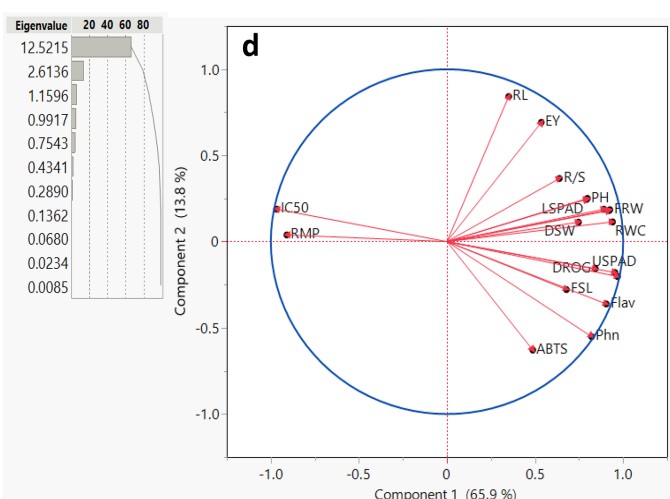

**Figure 5 Summary contribution by all principal components (PCs), a biplot between PC1 and PC2 displaying the distribution of parameters measured under different (0, 6, 12, and 18 dS/m NaCl) conditions.** Plant height (PH), fresh shoot length (FRL), root length (RL), and fresh root weight (FRW), fresh shoot weight (FSW), dried root weight (DRW), dried shoot weight (DSW), root/shoot index (%), drog, relative water content (RWC), relative membrane permeability (RMP), LSPAD, USPAD, extraction yield (EV), TAEC (DPPH), IC%50 (DPPH), phenol, flavonoid, and ABTS. (A) 0 dS/m, (B) 6 dS/m, (C) 12 dS/m, (D) 18 dS/m.

high IBA doses (2 g/L) cluster together. This shows they have a similar and significant impact on the plant's growth, physiology, and antioxidant parameters. These conditions generally lead to reduced growth, as well as decreased physiological and antioxidants activity, as shown by the dominance of blue color in this cluster. Combination of high salinity and high IBA exerts considerable stress on the plants, likely overwhelming their adaptive mechanisms.

Low salinity combined with low or no IBA (*e.g.*, 0 dS/m + 0 g/L IBA) forms a distinct cluster, these treatments produce mild or baseline responses in the plants. This cluster is marked by neutral or slightly red color, helping maintain normal physiological levels or

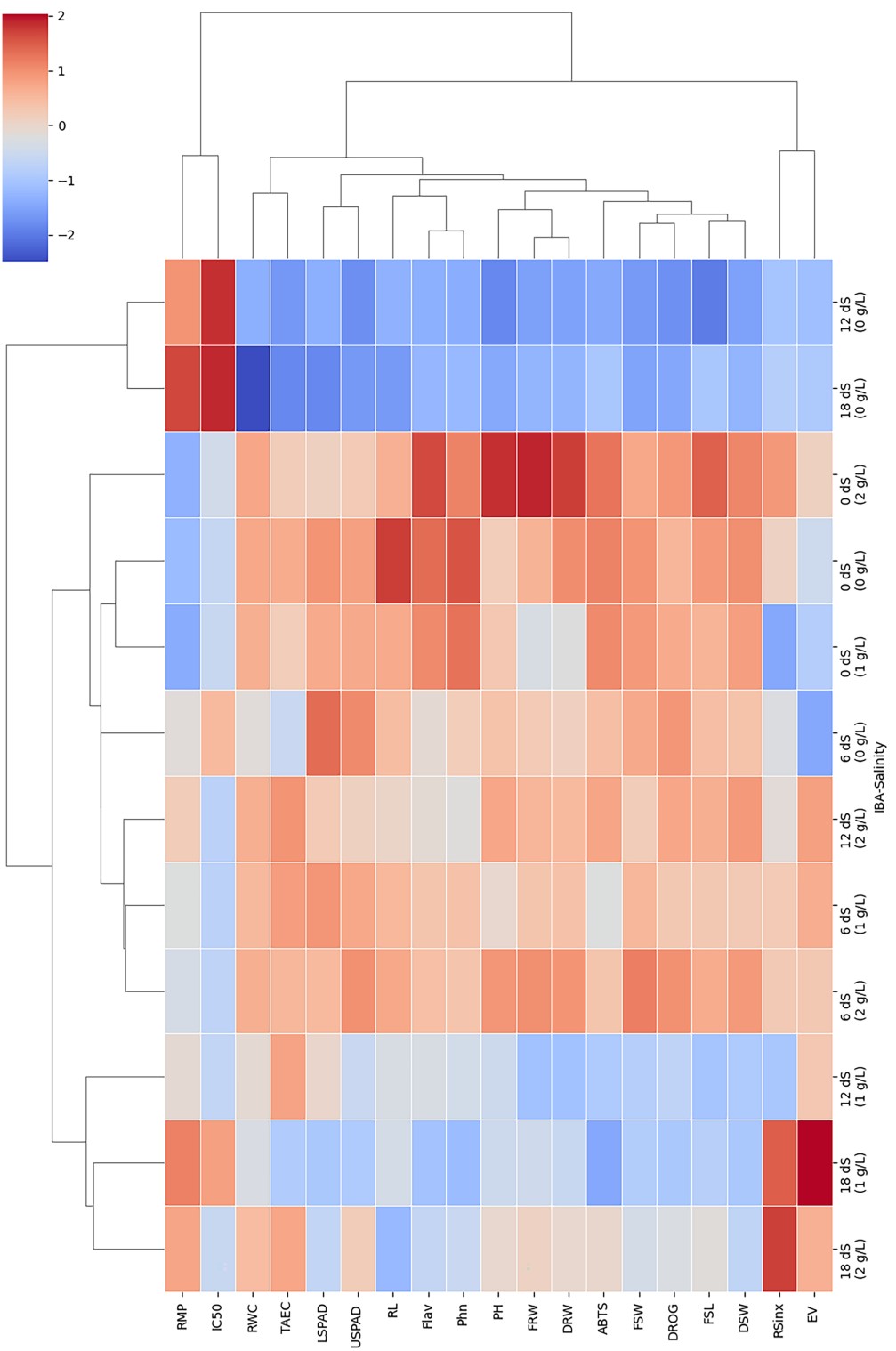

**Figure 6** The two-way hierarchical clustering heatmap shows the relation of varying IBA and salinity (NaCl) treatments on growth, physiological, and biochemical parameters. Plant height (PH), fresh shoot length (FRL), root length (RL), and fresh root weight (FRW), fresh shoot weight (FSW), dried root weight (DRW), dried shoot weight (DSW), root/shoot index (DRW/DSW), drog, relative membrane content (RMC), relative membrane permeability (RMP), LSPAD, USPAD, extraction yield (EV), TAEC

**Figure 6 (continued)**
(DPPH), IC%50 (DPPH), phenol, flavonoid, and ABTS. Red colour indicate an increase in response, blue colour a decrease, and neutral colour suggest minimal or no effect.

slightly enhancing them. The separation of this cluster from more stressful conditions indicates that the lack of both stress (low salinity) and growth-promoting hormones (low/no IBA) allows the plants to operate under near-optimal conditions, making this cluster a useful control or baseline reference. Moderate salinity combined with moderate IBA (*e.g.*, 6 dS/m + 1 g/L) forms a distinct cluster, typically associated with a mix of red and neutral color. This shows that moderate stress paired with moderate hormonal application leads to balanced plant responses, where some parameters show improvement while others remain stable.

RWC, RL, and FSW tend to cluster together; they are similarly affected by treatments, particularly those involving high salinity. These parameters are highly sensitive to salinity, with notable reductions observed under high-salinity conditions, especially when paired with high IBA doses. In contrast, antioxidant parameters, including TEAC and ABTS radical scavenging activity, form their own cluster, showing red color under moderate stress conditions. This clustering means these parameters may be co-regulated as part of the plant's defense mechanisms.

# DISCUSSION

Soil salinity is a significant environmental stressor impacts plant growth, leading to various agronomic, physiological and biochemical changes in plants (*Gupta & Huang, 2014*). These changes affect plant development, including germination, vegetative growth, and reproductive processes (*Singh et al., 2020*). In this study, these parameters were examined: plant height (cm), flower spike length (cm), drog yield (g/plant), fresh shoot weight (g/plant), dry shoot weight (g/plant), root length (cm), fresh root weight (g/plant), dry root weight (g/plant), root/shoot index, SPAD, relative membrane permeability (%), relative water content (%), DPPH (mg TEAC/g dw), ABTS (mg TEAC/g dw), flavonoid content (mg rutin/g dw), and phenolic content (mg GAE/g dw).

The results clearly show a general decline in growth, physiological, and antioxidant parameters as salinity stress increases from 0 to 18 dS m$^{-1}$. This decreasing trend in growth and physiological parameters is anticipated under conditions of increasing salinity stress (*Kulak, Gul & Sekeroglu, 2020*; *Hegazy, Awad & Abdelkader, 2021*; *Göçer et al., 2021*; *Sheikhalipour et al., 2024*).

On the other hand, other parameters of non-enzymatic antioxidant groups, including DPPH, ABTS, flavonoid, and phenolic content, normally show an increase under low stress conditions, then reverse into a decline trend as stress levels intensify; however, it is not the case in this study. This phenomenon has been documented in previous studies, where non-enzymatic antioxidants initially increase to counteract the harmful effects of reactive oxygen species (ROS) generated by environmental stress (*Ahmad et al., 2010*; *Amooaghaie et al., 2024*). Then, under more severe stress conditions plants are

overwhelmed, and the levels of these antioxidants rapidly decrease. This response can vary significantly among different plant species, with some showing a slight increase, others remaining stable under moderate stress, and some exhibiting an immediate decline, also depending on other factors such as stress type and stress intensity (*Mittler, 2002*; *Gill & Tuteja, 2010*).

However, in this study, a slight increase in SPAD values (lower and upper parts of leaves) was observed at 6 dS/m, while IC50 (mg dw/mg DPPH) and TEAC (mg/g dw) remained stable at this salinity level but showed a decrease at 12 dS/m and higher levels. This effect can be attributed to the decrease in osmotic potential in leaves caused by salinity, which reduces water content in leaf tissues. As a result, pigments like chlorophyll become more concentrated, leading to an increase in chlorophyll content per unit area (*Wang & Nii, 2000*). However, prolonged or severe exposure to stress ultimately impairs the photosynthetic system and causes a general decline in chlorophyll (*Heidari, Bandehagh & Toorchı, 2014*; *Li et al., 2020*; *Tan & Gören, 2024*).

In contrast, all other parameters showed a decrease with increasing stress. Notably, both RMP (%) and IC50 exhibited an increase under stress, but this behavior shows negative effect of the effects of salinity stress. While some parameters may initially resist or adapt to moderate salinity; they are eventually overwhelmed with increasing stress (*Ahmad et al., 2010*; *Alnusairi et al., 2021*). It was also mentioned by other researchers that SPAD values and antioxidant measures like IC50 and TEAC remain stable at moderate salinity levels but decline as salinity increases (*AbdElgawad et al., 2016*; *Younessi-Hamzekhanlu et al., 2021*).

On the other hand, the application of IBA doses before the rooting phase was observed to positively influence all parameters in this study. The beneficial effect of IBA can be related to its role in promoting the formation of a healthy and strong root system (*Abdel-Rahman, 2019*). This enhanced root system increases the plant's ability to absorb water and nutrients, thereby mitigating the negative impacts of salt stress (*Tognetti et al., 2010*; *Fattorini et al., 2017*). IBA acts as a precursor to indole-3-acetic acid (IAA), the main auxin in plants, which plays a role in root development and stress response. IBA promotes lateral root formation by sustaining a pool of IAA, thereby regulating root architecture and enhancing nutrient and water uptake under stress conditions. Studies indicate that IBA transport is mediated by proteins such as TOB1, linking it to cytokinin signaling, which balances root growth under stress (*Michniewicz et al., 2019*). Furthermore, IBA priming increases nitric oxide (NO) levels, a signaling molecule that facilitates root growth adaptation during osmotic stress (*Kolbert, Bartha & Erdei, 2008*). Thus, this indirectly helps to plant nutrient uptake, allowing the plant to continue growing even under stress conditions (*Šípošová et al., 2021*). Also, several studies have shown that IBA doses are used to enhance rooting in plants with poor root development, facilitating better root growth (*Maggio et al., 2010*; *Sağlam et al., 2014*; *Jiang et al., 2021*).

When examining the negative effects of salinity stress, it was found that IBA application, particularly at 2 g/L, significantly reduced the adverse effects of salinity stress on *Salvia fruticosa* seedlings across all measured parameters. It can be explained by feature of IBA that promotes root cell division and elongation, resulting in a deeper and more extensive root system. This enhanced root system increases the plant's ability to absorb water and

nutrients, thereby mitigating the negative impacts of salt stress (*Tognetti et al., 2010*; *Fattorini et al., 2017*).

The, PCA analysis across the different salinity conditions illustrates the dynamic relationship between various plant traits under the influence of salinity stress. Under control conditions (0 dS), the plants showed a more homogeneous interaction distribution among the parameters. However, as salinity increased, changes in the relationships between parameters were observed, with these interactions shifting in a more unified direction. The findings suggest that while certain traits such as root and shoot biomass are strongly linked to each other under control condition, salinity stress alters these relationships, and with stress-related parameters like RMP and IC50 becoming more prominent indicators, this shows cellular membrane injury due to salinity, and also reported by *Jamil et al. (2012)* The observed shifts in correlations and variability across increasing salinity levels show the complex interaction between parameters and salinity doses.

According to two-way hierarchical clustering heatmap (Fig. 6), strong positive correlations were observed between plant height (PH) and flower spike length (FSL), fresh root weight (FRW), and fresh shoot weight (FSW), this shows a relationship between plant height and overall plant growth. According to these results, it can be said that taller plants tend to be associate with greater biomass accumulation in both roots and shoots, as well as the development of longer flower spikes. Also, according to *Sun & Frelich*'s *(2011)* study, plant height is correlated with biomass accumulation, where taller plants had higher shoot and stem biomass, which in turn was associated with better reproductive output, including flower spikes.

Also, results show that fresh shoot weight (g/plant) has a strong relation with drog value, USPAD and flavonoid content. The plants that have a higher fresh shoot weight tend to have higher dry weight, better chlorophyll content in the upper leaves, and elevated flavonoid content levels. The close relationship between FSW and drog shows the importance of shoot biomass as a predictor of overall yield, while the correlations with USPAD and flavonoids highlight the role of shoot growth in supporting photosynthesis and secondary metabolite production. Also, TEAC shows strong positive correlations with Flavonoid content and Phenol content. IC50 measures the concentration required to inhibit 50% of the DPPH radical, with lower values indicating higher antioxidant activity. IC50 shows moderate to strong negative correlations with various parameters, especially flavonoid content and ABTS. This relationship means that higher levels of flavonoids and better ABTS antioxidant performance correspond to lower IC50 values, reflecting stronger antioxidant activity. The correlation matrix shows how interconnected various growth, stress, and antioxidant parameters are in determining plant production and growth. Strong positive correlations between growth parameters like PH, FSL, FRW, and FSW show their collective role in promoting plant development.

## CONCLUSIONS

This study shows the major effects of saline conditions as an environmental stressor on *Salvia fruticosa* seedlings. The results show a clear reduction in several growth,

physiological, and antioxidant parameters when salinity rises from 0 to 18 dS/m, except for DPPH (mg TEAC/g dw) and SPAD (chlorophyll content), which increased under moderate salinity stress but decreased under elevated stress levels. Notably, the application IBA, particularly at a concentration of 2 g/L, was effective for mitigating the negative effects of salinity stress. Across all measured parameters, IBA significantly reduced the adverse impacts of salinity on *Salvia fruticosa* seedlings. These findings are important because, compared to other stress mitigation methods that are commonly used and recommended, this application is not only practical but also seems effective even at high stress doses, as observed in this study. In the future, it is recommended to conduct studies comparing this method with other stress mitigation techniques, which will allow for a better evaluation of its effectiveness. The PCA provided further information on complex interactions between IBA treatment and salinity stress. Under normal conditions, traits like root and shoot biomass were closely linked, but these relationships shifted as salinity levels increased. Stress-related parameters, such as relative membrane permeability (RMP) and IC50 (DPPH), became more prominent under stress conditions. This shift emphasizes the importance of considering which traits are more sensitives when evaluating plant responses to environmental changes, as the interplay between these factors can offer a better understanding of plant resilience mechanisms. Additionally, the study shows the role of shoot biomass in overall yield, as fresh shoot weight was strongly correlated with drog value, upper SPAD (USPAD), and flavonoid content. This shows that plants with greater shoot biomass also tend to have higher drogs, better chlorophyll content, and increased flavonoid levels. As a results, a one-time IBA application on the plant cuttings before rooting stage can be a viable strategy for mitigating the adverse effects of salinity as a practical and effective method for enhancing salinity tolerance, especially in plants propagated vegetatively through cuttings. Therefore, for cultivation in soils with undesirable salinity levels or under saline irrigation conditions, this study demonstrates that *Salvia fruticosa* or similar plants that have vegetative propagation options benefit from the application of appropriate doses of IBA to the cuttings. By preparing the plants before exposure to salinity conditions, it is possible to enhance their resilience and ensure successful cultivation under adverse conditions.

### Funding
The authors received no funding for this work.

### Competing Interests
The authors declare that they have no competing interests.

### Author Contributions
- Uğur Tan conceived and designed the experiments, performed the experiments, analyzed the data, prepared figures and/or tables, authored or reviewed drafts of the article, and approved the final draft.

## Data Availability

The raw measurements are available in the Supplemental File.

## Supplemental Information

Supplemental information for this article can be found online at http://dx.doi.org/10.7717/peerj.18846#supplemental-information.

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
