# Peer review of "Application of indole-3-butyric acid (IBA) enhances agronomic, physiological and antioxidant traits of Salvia fruticosa under saline conditions: a practical approach"

_PeerJ, doi:10.7717/peerj.18846_

## Round 0.1 · original submission · Major Revisions

Dear Authors

The manuscript cannot be accepted for publication in its current form. It needs a major revision before publication. The authors are invited to revise the paper considering all the suggestions made by the reviewers. Please note that the requested changes are required for publication.

With Thanks

Reviewer 1 ·

Basic reporting

The manuscript titled "Application of Indole-3-Butyric Acid (IBA) Enhances Agronomic, Physiological, and Antioxidant Traits of Salvia fruticosa Under Saline Conditions: A Practical Approach" presents valuable insights into the use of IBA for improving plant resilience under saline stress. However, revisions are necessary to improve clarity and scientific rigor. The language contains many typos, grammatical errors, awkward phrasing, and instances of redundancy that affect the overall flow and readability. Improving sentence structure, eliminating repetitive expressions, and ensuring proper subject-verb agreement would enhance the clarity and professionalism of the text.
The introduction requires several improvements to strengthen its structure and focus: The transition between general information about Salvia fruticosa and salinity stress mitigation strategies is abrupt. After introducing S. fruticosa, then discuss why salinity stress is particularly relevant to this species, followed by a smoother transition into discussing mitigation strategies.
In Materials and Methods section, while the collection of cuttings is described, more information is needed about the criteria for selecting plants (e.g., age, health status) to ensure reproducibility. Were the cuttings from mature plants or a specific growth stage? The method for applying IBA (15-second immersion) is acceptable but could benefit from more precision. For example, was the temperature of the IBA solution controlled? The brief immersion time also raises the question of whether this is sufficient for proper absorption across different doses (especially 2000 ppm). The waiting period of 24 hours before watering to prevent washing out IBA is logical but lacks validation. Could this delay impact the viability of the cuttings or the initial rooting process? The study utilizes a two-factor randomized block design, which is appropriate for controlling variability. However, there is no rationale provided for the selection of salinity levels (0 dS m-1, 6 dS m-1, 12 dS m-1, and 18 dS m-1). These levels should be justified, particularly with reference to previous studies or field conditions typical for Salvia fruticosa. How was the total water volume determined across the salinity treatments, and how frequently were the salinity treatments applied? These details are critical for reproducibility. The antioxidant assays are well described, but the preparation of plant extracts needs further detail. What was the solvent used for the extraction, and how were the samples stored before analysis? It is also essential to specify the equipment used for absorbance measurements (e.g., brand and model of the spectrophotometer). The methods for flavonoid and phenolic content lack of description for how plant material was processed for extraction. Were leaves dried, ground, or used fresh?
The results show a significant positive impact of IBA application on plant growth, especially at higher concentrations (2000 ppm). For example, IBA substantially increases plant height, root length, and shoot biomass even under salinity stress, suggesting its role in enhancing stress tolerance. This is consistent with the role of IBA in promoting root growth and enhancing the absorption of water and nutrients, thereby improving plant vitality under stress conditions. The lack of significant differences in root length (RL) at certain IBA concentrations (e.g., no statistical difference between 1000 ppm and 2000 ppm) raises questions about IBA’s efficiency at higher dosages. It would be beneficial to explore if there is a saturation point where additional IBA offers no further benefit. It was observed key values are given in some cases but please add key values in terms of percentage increase or decrease for all the parameters under study and having promising results. Precise the results of PCA analysis please.
The discussion starts with a good contextual background regarding the negative impacts of soil salinity on plant growth, linking it to the relevant literature. However, the flow of information could be more structured. For instance, the discussion jumps between growth parameters and antioxidant activity without a clear thematic separation. The Discussion also moves quickly between different pieces of information. A more deliberate breakdown of each major finding, followed by a dedicated exploration of how each parameter (e.g., growth, antioxidant capacity) is impacted by salinity, would make the section more digestible for readers. Instead of simply stating that previous studies found similar trends, more comparative analysis could be helpful. For example, how exactly do the results align or diverge from similar studies? Are there any novel insights this study provides beyond confirming previous findings? Provide more explanation of why certain responses (e.g., antioxidant activity) occur and how IBA mitigates salinity stress on a physiological level. Regarding SPAD values, while a slight increase at 6 dS/m is noted, it’s important to clarify its in-depth logical context. Author could expand slightly on how IBA influences hormonal signaling under stress conditions. Is it primarily affecting root development, or does it have broader effects on growth-regulating hormones (e.g., auxins, gibberellins)? In case of PCA, a more explanation of how correlations shift under stress could make this section more accessible.
The conclusion section can be strengthened by providing a clearer and more concise summary of the key findings, eliminating repetitive statements. I should not be the repetition of results section, rather it must be generalized. The mechanistic insights into how IBA enhances root and shoot development under salinity stress should be elaborated, along with more confident phrasing. Finally, future research directions could be more specific, recommending comparisons with alternative methods and discussing potential limitations or scalability concerns of IBA application. By addressing these areas, the conclusion will provide a more comprehensive and impactful summary of the study's contributions.
Specific comments:
Drog (Line 24) give full form of this abbreviation.
Relative membrane content (RMC) (Line 24) what is it? Explain.
The introduction repeats the impact of salinity multiple times (lines 44, 47, 50), which leads to redundancy. Combining these statements into a single, concise explanation of salinity's effects would improve readability and focus.
“Additionally, plant growth promoting rhizobacteria (PGPR) (Cappellari et al., 2023).” What does the author want to convey here? (Lines 58-59)
The review of salinity mitigation strategies (lines 56–92) lacks a clear connection to the study's specific focus on Salvia fruticosa or the use of IBA. This section introduces a broad range of techniques (nanoparticles, seed priming, PGPR, etc.), but the relevance of these methods to the research question is unclear. There should be more focus on the specific challenge of using hormones like IBA and less on other strategies unless they are directly compared or contrasted with IBA. The introduction to Indole-3-Butyric Acid (IBA) appears late in the introduction (line 94) and is not well integrated with the prior discussion of salinity stress or other mitigation strategies. IBA is described as a “straightforward” method, but more scientific reasoning behind its selection, its mode of action, and how it compares to other approaches should be provided earlier in the introduction to make the study's rationale clearer.
“In contrast to other approaches, applying Indole-3-Butyric Acid (IBA) directly to cuttings during vegetative propagation offers a straightforward and potential method for enhancing salinity stress tolerance.” (Lines 94-96) Provide reference.
The objectives of the study are presented as questions (lines 102–105), which could be reformulated into clearer, declarative research aims. Instead of asking whether IBA can improve tolerance under salinity stress, the introduction should assert that the study will test this hypothesis. Additionally, the significance of this study, particularly the novel aspects of using IBA and focusing on S. fruticosa, should be emphasized more strongly.
The research gap is only mentioned briefly in the last few lines (lines 102-107), and it is not clearly stated how the use of IBA addresses a specific gap in salinity stress research. The gap in the literature on Salvia fruticosa is acknowledged, but the specific novelty of this study—whether it is the use of IBA, the focus on S. fruticosa, or the salinity conditions—needs to be more clearly highlighted earlier in the introduction.
The amount of salinity that will be created by 600 ml of water in a 3-liter pots with 1:1 peat/perlite mixture in greenhouse condition can be calculated using the formulas provided in Figure 1. (Lines 136-138) correct the sense of sentence.
The phrase “examined range of parameters” (lines 516–517) is vague and could benefit from clearer language. Instead, a specific statement about which key parameters were investigated would improve the clarity of the section.
While the response of non-enzymatic antioxidants is discussed (lines 524–530), more attention could be given to the mechanistic insights of why these antioxidants first increase and then decrease under severe stress. The explanation that this response is typical (based on Ahmad et al., 2010; Amooaghaie et al., 2024) is helpful, but the physiological reasoning behind the shifts could be further unpacked.
The role of IBA in mitigating salinity stress is discussed at length (lines 544–557), but the exact mechanisms by which IBA promotes salinity tolerance could be better explained. While the text briefly mentions that IBA enhances root development, a more detailed mechanistic explanation would elevate the discussion (e.g., by discussing how IBA influences root hormone signaling, nutrient uptake, or osmotic regulation under stress).
The PCA analysis (lines 558–568) is mentioned, but the discussion around it remains surface-level. A deeper analysis of the interaction between plant traits under salinity and IBA treatment would be beneficial. What do these shifts in correlations tell us about the plant's overall adaptive strategy? Why do some parameters become more prominent under stress? More explanation is required to provide insights into the practical implications of these shifts.
The results section mentions the correlation between growth parameters (e.g., PH, FSL, FRW) and stress indicators (e.g., RMP, IC50), but there could be more critical interpretation. Does this correlation provide new insight into the role of biomass in salinity tolerance? Also, the explanation that extraction yield (EY) and the root/shoot index do not correlate with other parameters (line 590–591) seems to suggest that these metrics are not useful in this context, but the discussion should attempt to explain why this is the case.
In some places, the discussion seems to describe trends rather than critically evaluate them. For example, there is room to discuss the biological significance of observed phenomena, such as why RMP and IC50 become more prominent indicators under stress (lines 561–563).
There are some awkward phrasings and grammatical issues. For example, the sentence "According to results it can be clearly a general decline..." (line 520) should be revised for clarity. Also, "exogenous hormone applications have been shown to have positive effects on salinity tolerance" (line 549) is too broad a statement and could be more specific about how IBA compares to other hormones under stress conditions.
Some sentences lack clarity, particularly the sentence “As results, the findings indicate that IBA application before can a viable strategy...” (lines 615–616). The phrase “before can a viable strategy” seems like an editing error and should be revised to “before exposure can be a viable strategy.” These small errors detract from the professionalism of the text.
In the footnotes of figures, it is essential to clarify the meaning of the alphabets on the bars of the graphs.
Add alphabets in Fig. 2h.
Avoid duplicating information in both tables and figures—retain either the table or the graph for clarity. ANOVA Tables can be added in supplementary material. Additionally, ensure that all figure and table captions are fully self-explanatory, providing the full forms of any abbreviations used within them. This will enhance readability and comprehension without requiring reference to other sections of the manuscript.

Experimental design

The experimental design outlined in the study seems valid. However, there are few concerns, the 1:1 peat-perlite mixture provides good aeration and water retention, but the use of such a substrate may differ from field conditions where salinity and soil texture interact. A mention of how these greenhouse results might extrapolate to field conditions would enhance the experimental relevance. More details are needed on how leachate collection and salinity measurement were standardized across replications and blocks.

Validity of the findings

The validity of the article's findings is strong due to a well-structured design and appropriate measurement techniques.

Reviewer 2 ·

Basic reporting

Need more references to explain the results in-depth.

Experimental design

OK

Validity of the findings

OK

Additional comments

peerj-reviewing-106273-v0

The report contains some useful results. However, it was too descriptive on results but with less in-depth discussion.

The IBA and salinity on the various properties of the Salvia fruticose should be discussed systematically, rather than independently presented in the current report. Their major effects should be reflected by modulating metabolomic pathways which caused the final property changes, especially that the leaves are made for herbal tea. The authors can search database like Web of Science with food metabolomics (Title) AND quality analysis (Title) to get reference to improve the discussion

Table 3: method for flavonoid content was not clearly stated in the method section. In addition, the related discussion on the flavonoid needs improvement. The parameter dietary flavonoids is important because they are related to health role which should be mentioned thus the readers can understand more why this parameter was tested. The authors can search database like Web of Science with dietary flavonoids (Title) AND health role (Title) to get reference to discuss it further.

Many chemical tests should be influenced by matrix effect. However, in this current report, the matrix compounds removal was quite successful thus the assay accuracy was pretty good. The in-depth discuss is needed on why in this report the matrix compounds removal was successful, resulted in good assay results. The authors are suggested to search database like Web of Science with matrix compounds removal (Title) AND assay (Title) to get reference to enhance the discussion.

Some format of units etc should be double checked. For instance, unit ppm should be changed to SI unit.

Figures 2-4: SD or SE should be added to each data point.

Reviewer 3 ·

Basic reporting

The “Application of Indole-3-Butyric Acid (IBA) enhances agronomic, physiological and antioxidant traits of Salvia fruticosa under saline conditions: a practical approach”, authored by Uğur Tan describes the use of indole-3-butyric acid as a protector of the physiological and antioxidant properties of Salvia fruticosa under stress conditions caused by excess salt in the soil. The protocol used seems appropriate to me according to the procedures described in the literature. However, experimental procedures need to be described in more detail and must always be accompanied by the reference that was used. I suggest carefully reviewing the entire text for typos. This work could be a relevant contribution to the scientific community. I recommend the acceptance of this manuscript after careful review and addition of the mentioned items.

Experimental design

The research agrees with the scope of the journal. However experimental procedures need to be better detailed so that they can be reproduced.
I would like the choice of exposure time of Salvia fruticosa to IBA to be better explained in the text. Is there any other study that guides this choice? A dose versus exposure time curve was performed. Please make this information clearer in the manuscript.

Validity of the findings

The results obtained are promising. However, the text must be reviewed before publication. The manuscript needs to be complemented.

---

## Round 0.2 · Major Revisions

Dear Author

The manuscript cannot be accepted for publication in its current form. It needs a major revision before publication. The author is invited to revise the paper considering all the suggestions made by the reviewers. Please note that the requested changes are required for publication.
With Thanks

Reviewer 1 ·

Basic reporting

The manuscript's language requires serious attention; there are instances of incorrect grammar usage, missing articles, and missing spaces between sentences.
Author use acronyms in multiple spaces without first giving their full form.
The author is repetitively giving the same information about salinity stress in different sentences, as Line No. 43-45 However, the cultivation of this valuable plant, like many others, is threatened by salinity a critical environmental challenge that severely limits plant growth and agricultural productivity (Atta et al., 2023).", Line 51-52 "Salinity is a critical environmental challenge that severely limits plant growth and agricultural productivity (Allakhverdiev et al., 2000). "
The novelty of the study is stated, but it is buried among excessive details on salinity mitigation.
The existing literature briefly mentions the gap but does not elaborate on it. Expanding on the lack of studies specifically targeting Salvia fruticosa and salinity stress.
The presentation of the objectives is abrupt and lacks a smooth transition from the preceding content.
There is unnecessary repetition of "Indole-3-Butyric Acid (IBA)" in the presentation of hypotheses.
Some references are incorrectly formatted or have inconsistent punctuation, such as "Rhaman et al., (2020)" at multiple places in the introduction and discussion section.
As previously suggested, when analyzing some of the most attractive results, consider adding a percentage increase or decrease in comparison to the control. It is strongly recommended to add values in terms of percentage.
The discussion is superficial; please provide a detailed, logical explanation of the obtained results in the context of current literature.
Why did SPAD values increase under stress? Why does chlorophyll content decline under high salinity stress? Discuss the main in-depth mechanism.
What does the term "dry plant separated from stems" mean?
At 0 dS/m (no salinity), the highest plant height is recorded at 54.00 cm. (remove zeros from value 54.00) and correct all such values in the results section by reducing their decimal values.
Correct Figure and Table numbers in the results section; the PCA analysis is presented in Fig. 5, not in Fig. 3. Similarly, the heatmap is presented in Figure 6, not in Figure 5.

Experimental design

The experimental design is well-structured and appropriate for addressing the research objectives.

Validity of the findings

Findings seems promising

Reviewer 2 ·

Basic reporting

Good

Experimental design

Good

Validity of the findings

Good

Additional comments

peerj-reviewing-106273-v1

The authors have addressed most of the questions quite well and the quality of the revised manuscript has been improved significantly. However, some few question needs further elaboration.

For the matrix effect, the response did not address the question well. The effect of matrix effect on chemical molecules analyses is critical. In this report, the results indicated the chemical analyses had good assay accuracy. However, the related discussion on reasons behind needs further improvement. Why did matrix compounds removal successful for achieving good assay accuracy? The authors can search topic ‘matrix compounds removal for assaying’ to get reference for improving discussion.

Reviewer 3 ·

Basic reporting

After reviewing the article again I was able to observe that the author made the suggested changes to the manuscript, meeting most of the requests. In this way I am in favor of accepting the manuscript as it is.

Experimental design

The experimental procedures were further detailed so that they could be reproduced.

Validity of the findings

The results obtained are promising and can be published in PeerJ.

---

## Round 0.3 · accepted · Accept

Dear Authors,

I am pleased to inform you that the manuscript has improved after the last revision and can be accepted for publication.

Congratulations on accepting your manuscript, and thank you for your interest in submitting your work to PeerJ.

With Thanks

Reviewer 1 ·

Basic reporting

The authors have made satisfactory corrections in the manuscript titled "Application of Indole-3-Butyric Acid (IBA) enhances agronomic, physiological, and antioxidant traits of Salvia fruticosa under saline conditions: a practical approach" . The authors have successfully clarified key aspects of their experimental design, addressed typographical and grammatical errors, and ensured that their statistical analyses are robust and appropriate. This study provides valuable insights into the practical applications of IBA in mitigating the adverse effects of salinity stress on Salvia fruticosa.

Experimental design

Experimental design seems appropriate

Validity of the findings

Findings are promising.

Reviewer 2 ·

Basic reporting

all revised and meet the requirement

Experimental design

all revised and meet the requirement

Validity of the findings

all revised and meet the requirement

Additional comments

No